# Measuring Heart Rate Variability in Patients Admitted with ST-Elevation Myocardial Infarction for the Prediction of Subsequent Cardiovascular Events: A Systematic Review

**DOI:** 10.3390/medicina57101021

**Published:** 2021-09-26

**Authors:** Crischentian Brinza, Mariana Floria, Adrian Covic, Alexandru Burlacu

**Affiliations:** 1Institute of Cardiovascular Diseases, 700503 Iasi, Romania; crischentian-branza@email.umfiasi.ro; 2Faculty of Medicine, University of Medicine and Pharmacy, 700115 Iasi, Romania; floria.mariana@umfiasi.ro; 3Military Emergency Clinical Hospital, 700483 Iasi, Romania; 4Nephrology Clinic, Dialysis and Renal Transplant Center, University Hospital, 700503 Iasi, Romania; 5Medical Sciences Academy, 700503 Iasi, Romania

**Keywords:** heart rate variability, ST-elevation myocardial infarction, prognosis, autonomic nervous system, risk assessment

## Abstract

*Background and objectives*: Ischemic heart disease represents the leading cause of death, emphasizing risk stratification and early therapeutic intervention. Heart rate variability (HRV), an indirect marker of autonomic nervous system activity, was investigated extensively as a risk factor for adverse cardiovascular events following acute myocardial infarction. Thus, we systematically reviewed the literature to investigate the association of HRV parameters with mortality and adverse cardiovascular events in patients presenting with ST-elevation myocardial infarction (STEMI). *Materials and methods*: Following the search process in the MEDLINE (PubMed), Embase, and Cochrane databases, nine studies were included in the final analysis. *Results*: Lower time-domain HRV parameters and a higher ratio between power in the low-frequency (LF) band and power in the high-frequency (HF) band (LF/HF) were associated with higher all-cause mortality during follow-up, even in patients treated mainly with percutaneous coronary interventions (PCI). Although most studies measured HRV on 24 h ECG recordings, short- and ultra-short-term measures (1 min and 10 s, respectively) were also associated with an increased risk of all-cause mortality. Although data were discrepant, some studies found an association between HRV and cardiac mortality, reinfarction, and other major adverse cardiovascular events. *Conclusions*: In conclusion, HRV measurement in patients with STEMI could bring crucial prognostic information, as it was associated with an increased risk of all-cause mortality documented in clinical studies. More and larger clinical trials are required to validate these findings in contemporary patients with STEMI in the context of the new generation of drug-eluting stents and current antithrombotic and risk-modifying therapies.

## 1. Introduction

Heart rate variability (HRV) can be briefly characterized as the physiological variation of RR intervals between successive depolarizations of the sinus node. HRV can be considered an indirect marker of autonomic nervous system activity, which allows a relatively superficial insight into the complex structure of heart–brain connection [1].

HRV reflects the psycho-emotional status linked to different disorders such as stress or anxiety [2,3]. Interestingly, HRV represents not only a consequence of the action of various factors on the brain–heart connection, but it also can modulate the nervous system function. HRV biofeedback represents a respiratory training intervention that modifies the activity of different brain regions with beneficial effects documented in clinical studies on autonomic markers, stress levels, and anxiety, even in healthy individuals [4,5,6,7].

However, HRV constitutes more than a simple variation of RR intervals, as it can be expressed by different parameters using time-domain, frequency-domain, or nonlinear methods of measurement [8]. Four time-domain parameters were recommended for HRV assessment by the Task Force of the European Society of Cardiology and the North American Society of Pacing and Electrophysiology, namely standard deviation of all NN intervals (SDNN), HRV triangular index, the standard deviation of the average NN interval over short time divisions (SDANN), and the square root of the mean squared differences of consecutive NN intervals (RMSSD). Low-frequency power (LF), high-frequency power (HF), and the LF/HF ratio represent the most used frequency-derived parameters endorsed by the guidelines [9].

One of the earliest clinical applications of HRV measurement is represented by mortality risk stratification in patients with acute myocardial infarction (AMI). One study published in 1987 involving 808 patients with AMI observed that an SDNN less than 50 ms was associated with a 4-fold increase in mortality risk compared to patients with an SDNN more than 100 ms (34% vs. 9%, respectively) during a 4-year follow-up period. The usefulness of HRV for mortality stratification remained statistically significant after multivariable analysis. Moreover, HRV was associated with left ventricular function and segmental contraction abnormalities, as well as clinical and radiographic signs of pulmonary congestion [10].

HRV measurement is also helpful for mortality risk assessment in patients with end-stage kidney disease and chronic hemodialysis. A meta-analysis published in 2021 observed that both time- and frequency-domain parameters (SDNN, SDANN, and LF/HF ratio) were associated with increased all-cause death and cardiovascular mortality in hemodialysis patients [11].

Although most studies investigated HRV measured by 24 h electrocardiographic (ECG) recordings, many parameters were also validated in case of shorter time intervals. In one study that recruited 900 patients without coronary heart disease, HRV was measured using 2 min ECG strips. Patients with the lowest SDNN values had an increased risk of all-cause death and cardiovascular mortality. Other time-domain parameters such as RMSSD and pNN50 were also linked to increased mortality [12].

A recent meta-analysis involving patients with known cardiovascular diseases documented that patients with lower HRV exhibited a higher risk of all-cause death and cardiovascular events, an effect maintained in a subgroup analysis in participants with AMI. Although this meta-analysis also included studies that recruited patients with AMI, it was not focused on the specific population presenting with ST-elevation myocardial infarction (STEMI) [13].

We aimed to systematically review the literature to investigate the association of HRV parameters with mortality and adverse cardiovascular events in patients presenting with STEMI.

## 2. Materials and Methods

The present systematic review was conducted according to the updated Preferred Reporting Items for Systematic Review and Meta-Analyses (PRISMA) checklist [14].

### 2.1. Data Sources and Search Strategy

We performed a literature search in the MEDLINE (PubMed), Embase, and Cochrane databases from the inception until 20 August 2021, without using time intervals or language filters. Whenever possible, the search was restricted to studies that enrolled humans. As recommended by the PRISMA checklist, to find additional studies, we also screened cited articles, the Google Scholar search engine, and the ClinicalTrials.gov database of clinical trials. The following MeSH terms and keywords were used to retrieve references from the mentioned databases: “heart rate variability”, “myocardial infarction”, “ST-elevation myocardial infarction”, “cardiovascular events”, “risk assessment”, and “mortality”. The search process is provided in Appendix A.

### 2.2. Eligibility Criteria and Outcomes

Obtained citations from prespecified databases were considered for inclusion in the present systematic review if they met several criteria: (1) humans ≥18 years old were included for the analysis; (2) patients presenting with STEMI were enrolled; (3) standard HRV parameters endorsed by the guidelines were investigated [9]; (4) HRV was measured during the hospitalization for STEMI or at a distance from the index event; and (5) original data regarding the association between HRV parameters and the risk of adverse cardiovascular events or mortality in patients presenting with STEMI were reported. In addition, studies available only in abstracts, editorials, letters, case reports, conference papers, unpublished data, and meta-analyses were excluded from the analysis. In addition, studies were excluded due to the inability to extract data. Two independent investigators analyzed the eligibility criteria, and disagreements were solved by consensus.

### 2.3. Data Collection

Two independent investigators extracted the following data: the first author, publication year, number of patients included, their age, HRV parameters evaluated, clinical setting, follow-up period, reported outcomes, timing, and methods for HRV measurements. When available, data were presented as numbers, percentages, median or mean values, hazard ratio (HR), and odds ratio (OR), and the corresponding confidence intervals (*p*-values).

### 2.4. Quality Assessment

The quality of observational studies without a control group was appraised using a tool provided by the National Institutes of Health (NIH). It contained 14 key questions that guided the overall evaluation of the studies’ quality [15].

## 3. Results

Our search in the databases mentioned above and in additional sources identified 2995 references. Subsequently, duplicate citations and those based on title or abstract were excluded, thus leaving 124 studies for eligibility assessment. After full-text examination, nine studies were included in the present systematic review. The search process is detailed in Appendix A.

General characteristics of clinical studies, including the number of patients and their age, HRV parameters evaluated, timing and methods of measurement, and outcomes investigated, are provided in Appendix A. In addition, results reported in studies are illustrated in Table 1.

All analyzed studies had an observational, nonrandomized design [16,17,18,19,20,21,22,23,24]. Most of the studies (*n* = 5) investigated patients prospectively [16,17,20,21,24], while three studies were retrospective [19,22,23], and one was cross-sectional [18]. In the majority of cases, HRV was measured based on 24 h ECG recordings [16,17,18,19,20,21,22], while two studies measured short or ultra-short time recordings (10 s and 1 min) [23,24]. Concerning myocardial reperfusion therapy, four studies enrolled mainly patients treated with primary percutaneous coronary intervention (pPCI) [19,20,21,22]. As all included studies had an observational design, the quality was judged to be fair, as guided by the NIH assessment tool (Appendix A).

Concerning all-cause mortality, clinical studies found an association with both time- and frequency-domain HRV parameters. Balanescu et al. documented significantly lower SDNN and RMSSD values (*p* < 0.001) in patients who died at 1-year follow-up compared to those who survived. In addition, deceased patients exhibited higher LF and lower HF values (*p* < 0.001 for both), most likely denoting an increased sympathetic tone. Importantly, 80.9% of patients who died had an LF/HF ratio value greater than 2. Moreover, an LF/HF ratio > 2 displayed good sensitivity, specificity, and a negative predictive value, but a low positive predictive value (80%, 83%, 96%, and 45%, respectively), in comparison to SDNN < 50 ms, which had better specificity and a positive prediction value (98% and 85%, respectively), but lower sensitivity and a negative prediction value (58% and 93%, respectively) [16].

Boskovic et al. observed similar results, with reduced time-domain parameters (SDNN, mean RR interval, RRmax–RRmin) in patients who died in the 1-year follow-up, providing additional prognostic value to classic risk markers [17]. Compostella et al. found that patients with the lowest SDNN values had a higher risk of all-cause mortality (*p* = 0.010), even in patients with a high successful percutaneous coronary intervention (PCI) rate (94%), an effect that was not maintained in the case of RMSSD [19]. Ablonskyte-Dudoniene et al. revealed that patients with a reduced RMSSD had an almost 10-fold higher 1-year mortality risk, and those with low SDNN values had a 4-fold higher 5-year mortality risk [21].

In contrast to the studies mentioned earlier, when HRV was measured on 24 h ECG recordings, Karp et al. observed that measurement during an ultra-short time (10 s) also had important prognostic implications. Patients with low SDNN values at admission (before reperfusion) exhibited an almost 3-fold higher risk of 2-year mortality, even after multivariate analysis [23]. Katz et al. also investigated the utility of HRV measured on a short ECG strip (1 min), but during deep breathing. The authors observed that all patients who died during follow-up had reduced SDANN values [24].

In comparison with all-cause mortality, data regarding cardiac mortality were discrepant across studies. Balanescu et al. observed that patients who experienced sudden cardiac death within one year of follow-up had lower RMSSD, SDNN, and HF values, but a higher LF and LF/HF ratio, thus reflecting a high sympathetic tone in deceased patients. As prognostic markers for sudden cardiac death, an LF/HF ratio > 2 showed an excellent negative predictive value (98%), but with lower sensitivity (81%), specificity (74%), and positive predictive value (14%). A similar negative predictive value was found in the case of patients with SDNN less than 50 ms (97%), with reduced sensitivity (63%), specificity (87%), and poor positive predictive value (21%) [16].

Although depressed SDNN values had an independent predictive value for cardiac mortality (*p* < 0.01), Boskovic et al., observed that it was not associated with an increased risk of sudden cardiac death (*p* > 0.05) [17]. Compostella et al. did not find an association between low SDNN values and cardiac mortality (*p* = 0.055), even though it was statistically significant in the case of all-cause mortality [19]. Nevertheless, Ablonskyte-Dudoniene et al. documented an almost 10-fold higher risk of 5-year cardiac mortality in patients with low SDANN thresholds treated mainly with PCI [21].

Data regarding the correlation between HRV and the risk of major clinical events (MCE) were also discrepant in the literature. Coviello et al. observed that patients with MCE (death, new AMI) exhibited reduced SDNN and HF values, while LF was higher when compared to patients without MCE. After multivariate analysis, both predischarge SDNN and LF were associated with a significantly higher risk of MCE and reinfarction [20]. SDNN was also linked to a 4-fold higher risk of nonfatal MI, and an almost 5-fold risk of revascularization at 5-year follow-up [21]. In addition, HRV (SDNN, mean RR interval) was correlated with a higher thrombolysis in myocardial infarction (TIMI) score [18]. However, Compostella et al. found different results, as SDNN values were not linked to an increase in MCE risk (*p* = 0.367), and were associated only with all-cause death [19]. Moreover, Karp et al. did not report any difference in reinfarction risk, coronary artery bypass graft (CABG) surgery incidence, and risk of hospitalizations for cardiac causes regarding SDNN values at admission [23].

## 4. Discussion

To our knowledge, our systematic literature review was the first one focused on the utility of HRV measurement in patients admitted with STEMI.

Published guidelines regarding HRV measurement standards have recognized since 1996 the importance of HRV assessment in various pathological conditions, including hypertension, congestive heart failure, MI, cardiac arrest, and supraventricular and ventricular arrhythmias [9].

Nevertheless, HRV seemed to be forgotten in the next decade, as it was not adopted for risk stratification by any guidelines on STEMI, non-ST segment elevation myocardial infarction (NSTEMI), or chronic coronary syndromes [25,26,27,28,29]. The enthusiasm for HRV evaluation was raised once more advanced technology became available, including different sensors and wearable devices, allowing a more straightforward measurement of various HRV parameters, even in a contactless way [30,31,32].

Ischemic heart disease represents the leading cause of death, emphasizing the importance of risk stratification and early therapeutic intervention [33]. A recently published meta-analysis involving patients with coronary artery disease revealed that HRV parameters were associated with an increased risk of overall mortality, including sudden cardiac death, sudden cardiac arrest, nonsudden cardiac death, and noncardiac death (*p* < 0.001). In addition, diminished SDNN values were linked to other established risk factors, such as reduced left ventricular ejection fraction [34]. The great merit of this study was that it consolidated and confirmed the hypothesis of the (general) correlation of ischemic coronary heart disease and HRV, but it did not mention specifically acute coronary events or STEMI.

Moreover, HRV could be used to predict cardiovascular events even in a healthy population. One meta-analysis involving patients without known cardiovascular disease documented impressive results, as reduced HRV values were associated with a 40% higher risk of a first cardiovascular event. This meta-analysis included studies that measured HRV on different duration strips, from 10 s to 24 h evaluation [35]. Thus, HRV assessment could provide important prognostic information for cardiovascular risk stratification.

Mortality in patients with STEMI remains high, although it decreased significantly once PCI became available and more invasive secondary prevention measures were implemented in daily clinical practice. In one study that enrolled patients with STEMI who underwent primary PCI, the authors observed a 13.7% mortality rate during a mean follow-up of 3.5 years [36]. Therefore, risk stratification still represents a crucial step for each patient presenting with STEMI.

However, none of the risk scores available confers a perfect prediction power for adverse events, including TIMI, Global Registry of Acute Coronary Events (GRACE), Controlled Abciximab and Device Investigation to Lower Late Angioplasty Complications (CADILLAC), and Primary Angioplasty in Myocardial Infarction (PAMI) scores [37,38]. Inclusion of a supplementary marker such as HRV in traditional risk scores may improve the detection of high-risk patients, thus ensuring an early therapeutic intervention. Furthermore, HRV could be incorporated in future risk stratification computational models based on artificial intelligence to increase accuracy.

Overall, HRV appeared to be a helpful marker in stratifying the risk of all-cause mortality in patients with STEMI. In particular, lower SDNN and RMSSD values and higher LF/HF ratios were observed in high-risk patients [16,17,19,21,22,23,24]. However, these results should be interpreted cautiously, as many factors can impact HRV measurements: nonmodifiable factors (age, gender), environmental factors, physiological and pathological factors, lifestyle, and neuropsychological factors [39]. Although an increased LF/HF ratio was associated with worse clinical outcomes, a lower LF/HF ratio was observed in patients in prolonged lying positions due to various pathological conditions (*p* < 0.001) [40]. This could be of particular interest in patients with STEMI and low physical activity limited by heart failure symptoms.

Moreover, long-term treatment with beta-blockers recommended in patients presenting with STEMI can also impact HRV assessment. Beta-blockers can increase time- and frequency-domain parameters (HF) by reducing the activity of the sympathetic autonomic nervous system, thus improving clinical outcomes [17,41,42]. However, there are limited data concerning patients whose HRV parameters remained unchanged under treatment with beta-blockers. Thus, more clinical trials are required to elucidate if this could be a sign of a poor prognosis. Consequently, awareness is required when evaluating HRV parameters in patients treated with beta-blockers, as they could have higher values.

Although 24 h ECG monitoring was used in most of the clinical studies analyzed, there was also supportive evidence for the utility of short and ultra-short time-recording evaluations (<5 min) [23,24]. HRV measured even on a 10 s ECG strip before revascularization therapy was associated with significantly increased 2-year mortality [23].

Nonetheless, studies analyzed in the present systematic review were observational, with different PCI rates than myocardial reperfusion treatment, limiting the results in the contemporary cohort of patients with STEMI in the context of the new generation of drug-eluting stents and current antithrombotic and risk-modifying therapies. Hence, large clinical trials are required to confirm the utility of HRV assessment for risk stratification in contemporary patients with STEMI.

## 5. Conclusions

HRV facile measurement in patients with STEMI can provide major prognostic information, as it is associated with an increased risk of all-cause mortality documented in clinical studies. Currently, the available evidence supports the integration of HRV parameters in future prediction models to identify high-risk patients who would benefit from more aggressive or invasive preventive measures. Although data were discrepant, some studies found an association between HRV and cardiac mortality, reinfarction, and other major adverse cardiovascular events. More and larger clinical trials are required to validate these findings in contemporary patients with STEMI.

## Figures and Tables

**Table 1 medicina-57-01021-t001:** The results reported in clinical studies.

Author, Year	Outcomes	Parameters	Results	
Balanescu et al., 2004	1-year total mortality		Nonsurvivors vs. survivors	
		RMSSD, ms	9.6 ± 3.1 vs. 32.6 ± 10.9	*p* < 0.001
		SDNN, ms	37 ± 10.3 vs. 144 ± 41	*p* < 0.001
		LF, ms^2^	1409 ± 143 vs. 1241 ± 131	*p* < 0.001
		HF, ms^2^	443 ± 105 vs. 883 ± 184	*p* < 0.001
		LF/HF > 2	80.9% of patients who died vs. 8.1% patients who survived	*p* < 0.0001
	Sudden cardiac death at 1-year follow-up		Nonsurvivors vs. survivors	
		RMSSD, ms	9.3 ± 2.5 vs. 30.2 ± 2.5	*p* < 0.001
		SDNN, ms	36.7 ± 10 vs. 133 ± 51	*p* < 0.001
		LF, ms^2^	1382 ± 152 vs. 1260 ± 142	*p* < 0.001
		HF, ms^2^	451 ± 112 vs. 836 ± 223	*p* < 0.001
		LF/HF > 2	81% of deceased patients vs. 15.6% survivors	*p* < 0.0001
Boskovic et al., 2014	All-cause mortality at 1 year		Nonsurvivors vs. survivors	
		SDNN, ms	60.55 ± 12.84 vs. 98.38 ± 28.21	*p* < 0.001
		Mean RR interval, ms	695.82 ± 65.87 vs. 840.07 ± 93.97	*p* < 0.001
		RRmax–RRmin, ms	454.36 ± 111.00 vs. 600.99 ± 168.72	*p* = 0.006
Chakrovortty et al., 2011	Correlation between HRV and TIMI risk score		Low-risk group (TIMI 0–2) vs. intermediate-risk group (TIMI 3–7) vs. high-risk group (TIMI ≥ 8)	
		SDNN, ms	120.0 ± 19.8 vs. 71.0 ± 20.5 vs. 40.9 ± 6.4	*p* < 0.001
		Mean RR interval, ms	836.8 ± 121.0 vs. 776.7 ± 130.3 vs. 649.7 ± 75.5	*p* < 0.001
Compostella et al., 2017	Major clinical events	SDNN, ms	10 events from 52 patients in the lowest SDNN quartile vs. 21 events from 150 patients in the other quartiles (X^2^ = 0.813)	*p* = 0.367
	All-cause death	SDNN, ms	3 of 4 deaths occurred in the lowest SDNN quartile	*p* = 0.010
	Cardiac mortality	SDNN, ms	2 of 3 deaths occurred in the lowest SDNN quartile	*p* = 0.055
Coviello et al., 2013	Major clinical events		Predischarge HRV parameters, univariate analysis	
		Mean RR, ms	HR 0.99 (95% CI, 0.98–1.00)	*p* = 0.06
		SDNN, ms	HR 0.97 (95% CI, 0.95–0.99)	*p* = 0.009
		SDNNi, ms	HR 0.97 (95% CI, 0.93–1.00)	*p* = 0.08
		VLF, ms	HR 0.94 (95% CI, 0.89–0.98)	*p* = 0.007
		LF, ms	HR 0.88 (95% CI, 0.80–0.96)	*p* = 0.006
		HF, ms	HR 0.88 (95% CI, 0.77–1.00)	*p* = 0.05
			HRV parameters at 6 months (no MCE vs. MCE)	
		Mean RR, ms	916.4 ± 122.6 vs. 867.6 ± 68.6	*p* = 0.29
		SDNN, ms	139.1 ± 38.2 vs. 141.4 ± 41.4	*p* = 0.88
		SDNNi, ms	58.6 ± 21.6 vs. 52.7 ± 16.5	*p* = 0.48
		VLF, ms	53.8 ± 39.9 vs. 43.9 ± 12.1	*p* = 0.11
		LF, ms	26.1 ± 10.5 vs. 21.9 ± 7.0	*p* = 0.30
		HF, ms	18.2 ± 12.1 vs. 14.2 ± 5.8	*p* = 0.18
			Predischarge HRV parameters, multivariate analysis	
		SDNN, ms	HR 0.97 (95% CI, 0.952–0.996)	*p* = 0.02
		LF, ms	HR 0.90 (95% CI, 0.819–0.994)	*p* = 0.04
	Reinfarction		Predischarge HRV parameters, multivariate analysis	
		SDNN, ms	HR 0.96 (95% CI, 0.936–0.991)	*p* = 0.009
		LF, ms	HR 0.90 (95% CI, 0.81–1.009)	*p* = 0.07
Ablonskyte-Dudoniene et al., 2012	1-year mortality	RMSSD (≤20.9 ms)	HR 9.69 (95% CI, 1.88–49.95)	*p* = 0.007
	5-year all-cause mortality	SDNN (≤100.42 ms)	HR 4.36 (95% CI, 1.68–11.35)	*p* = 0.003
	5-year cardiac mortality	SDANN (≤85.41 ms)	HR 9.65 (95% CI, 1.27–73.4)	*p* = 0.029
	Recurrent nonfatal MI	SDNN (≤123.43 ms)	HR 4.1 (95% CI, 1.54–11.32)	*p* = 0.005
Erdogan et al., 2008	All-cause mortality	SDNN, ms	102 ± 39 (survivors) vs. 81 ± 33 (nonsurvivors)	*p* = 0.02
			OR 0.95 (95% CI, 0.95–1)–multivariate analysis	*p* = 0.1
	4-year survival	SDNN, ms	80% (SDNN < 50) vs. 92% (SDNN > 50)	*p* < 0.001
Karp et al., 2009	2-year mortality	SDNN, ms (admission)	OR 2.9 (95% CI, 1.12–7.56)	*p* = 0.028
	Reinfarction	SDNN, ms (admission)	3.1 ± 0.9 (reinfarction) vs. 3.0 ± 0.9 (no reinfarction)	*p* = 0.7
Katz et al., 1999	All-cause mortality	RRmax–RRmin, beats/min	OR 1.38 (95% CI, 1.13–1.63)	*p* = 0.028
		SDANN, ms	All patients who died (*n* = 10) had SDANN < 50 ms	

HF = power in high-frequency range; HRV = heart rate variability; LF = power in low-frequency range; RMSSD = the square root of the mean of the sum of the squares of differences between adjacent NN intervals; RRmax–RRmin = difference between the longest RR interval and the shortest RR interval; SDANN = standard deviation of the averages of NN intervals in all 5 min segments of the entire recording; SDNN = standard deviation of all NN intervals; SDNNi = mean of the standard deviations of all NN intervals for all 5 min segments of the entire recording; TIMI = thrombolysis in myocardial infarction; VLF = power in very-low-frequency range.

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
