# Peer review of "Measuring Heart Rate Variability in Patients Admitted with ST-Elevation Myocardial Infarction for the Prediction of Subsequent Cardiovascular Events: A Systematic Review"

_medicina, 2021, doi:10.3390/medicina57101021_

Round 1

Reviewer 1 Report

The review paper " measuring heart rate variability...." is a well-written meta-analysis and review paper, that gives a very good overview over the field of HRV as a  prognostic marker in patients with acute coronary artery disease. I did not find any flaws or misconclusions. It is good to read.

The chapter "Materials and methods" is described extensively. This is kind of hard to work yourself through this chapter since it does not really add to the subject itself. The figure 1 and the table 1 may not be really necessary for this review, please think about to put those into a supplemental part. You may leave in table 2, that describes the studies that were included in this meta analysis. 

In the discussion the authors describe the altered HRV unter chronih ß-blockade, which is a therapeutic principle in patients after acute myocardial infarction. How do the authors judge the reduced HRV under ß-blockade? Is this also a sign for a poor outcome? Or does the diagnosis "Reduced HRV" not correlate with the outcome in the case of medical treatment with ß-blockers? The authors may discuss this in one or two additinal sentences.

However, the paper is well-writtenMy comments above are just suggestions

Reviewer 2 Report

Measuring Heart Rate Variability in patients admitted with ST elevation myocardial infarction for the prediction of subsequent cardiovascular events: a systematic review

This systematic review highlights the importance of HRV in mortality and adverse cardiovascular events in patients presenting STEMI. The study is overall well conducted and thought of; however, the limitations are critical. Hence first, the data form the literary search is to discrepant leading to a huge gap in bias (type of MI treatment, therapy, timing, heart rate registration criteria, ecc.). A good idea could have been including STEMI and nSTEMI patients to assess each MI type and highlights HRV differences. I would suggest doing original research on this topic instead of reviewing it. Tables should be made fit for reading, in particular colon 6 (setting) of table 1 is almost impossible to read. Finally, only minor English editing in required.

Round 2

Reviewer 2 Report

The authors have made the adjustments as requested.